# How Effective Are Push–Pull Semiochemicals as Deterrents for Bark Beetles? A Global Meta-Analysis of Thirty Years of Research

**DOI:** 10.3390/insects14100812

**Published:** 2023-10-13

**Authors:** Somia Afzal, Helen F. Nahrung, Simon A. Lawson, Richard Andrew Hayes

**Affiliations:** Forest Industries Research Centre, School of Science, Technology and Engineering, University of the Sunshine Coast, Ecosciences Precinct, Dutton Park, Brisbane, QLD 4102, Australia; hnahrung@usc.edu.au (H.F.N.); slawson@usc.edu.au (S.A.L.); rhayes@usc.edu.au (R.A.H.)

**Keywords:** *Dendroctonus*, *Ips*, attractant, repellent, forest entomology, conifer

## Abstract

**Simple Summary:**

Bark beetles are global forest pests that feed and breed in the inner bark of trees and may transfer symbiotic fungi that cause blue staining and reduce the commercial value of timber. In the present study, we examined the effects of semiochemical treatments on deterring bark beetle populations. This population reduction was measured by a reduction in the attraction to lure/trap catches, tree mortality, and attacks on trees. Based on the analysis of 863 experiments, we found that semiochemical treatments effectively reduce the bark beetle population and, therefore, represent an eco-friendly technique for forest protection.

**Abstract:**

Bark beetles (Coleoptera: Curculionidae: Scolytinae) are among the most damaging tree pests globally. Rising temperatures, drought, fire, storms, cyclones, and poor forest management cause stress and loss of vigour in trees, and these conditions favour bark beetle outbreaks. While research has been conducted on push–pull strategies to deter bark beetles, using attractive and deterrent semiochemicals, the potential of this strategy to reduce bark beetle populations, particularly in the genera *Dendroctonus* and *Ips*, remains uncertain. Here, we conducted a global meta-analysis of 52 research articles to quantify the effects of semiochemical treatments on managing different species of *Dendroctonus* and *Ips* for forest protection. Based on this analysis, we found that push–pull semiochemicals can significantly reduce *Dendroctonus* and *Ips* populations measured by a reduction in the attraction to lure/trap catches, tree mortality, and attacks on trees. The overall efficacy of the push–pull semiochemical treatment shows a 66% reduction for *Ips* compared to control and a 54% reduction compared to control for *Dendroctonus*, while, at the species level, there was a 69% reduction for *Dendroctonus ponderosae* (Hopkins) and a 94% reduction in *Ips perturbatus* (Eichhoff), and a 93% reduction in *Ips latidens* (LeConte). Interestingly, among different treatment sources, the efficacy of conspecific semiochemicals in combination with heterospecific semiochemicals and non-host volatiles showed a 92% reduction in *Dendroctonus* spp., and conspecific semiochemicals in combination with non-host volatiles showed a 77% significant reduction in *Ips* spp., while the efficacy of heterospecific semiochemicals in reducing *Ips* population was about 69%, and 20% in *Dendroctonus*. Among different ecological regions, the use of a push–pull strategy showed a 70% reduction in *Dendroctonus* in central-west North America, and *Ips* showed a 75% reduction in southwest North America. Our results demonstrate that semiochemical-based push–pull techniques have the potential to reduce *Dendroctonus* and *Ips* bark beetle populations. Furthermore, based on our analysis, the efficacy of such eco-friendly interventions could be further improved and provide a good tool for forest managers to control these pests, at least under some circumstances.

## 1. Introduction

Bark beetles (Coleoptera: Curculionidae: Scolytinae) can cause significant economic and environmental impacts on plantations and native forests [1]. These beetles spend much of their life in the host tree and have symbiotic associations with bacteria, fungi, viruses, and other arthropods [2]. Symbiotic fungi transferred to trees by bark beetles can block conducting tissues [3] and reduce the commercial value of timber by causing blue staining. In outbreak conditions, they can reduce harvest yield, ecological diversity, and tree carbon sequestration [4,5].

It is estimated that the western United States of America (USA) experiences an annual US$2 billion loss due to bark beetle infestations [6]. The Czech Republic experienced €260 million in economic losses during the 2018–2019 outbreak of *Ips typographus* (L.), while *Dendroctonus frontalis* Zimmerman caused a loss of about US$1 billion over the last 28 years in South Carolina alone [3]. During 1989–2004, outbreaks of *Dendroctonus rufipennis* (Kirby) killed 30 million trees annually in Alaska [6], while outbreaks of *D. ponderosae* in British Columbia increased carbon emissions by 250 megatons between 2000–2020 and tree mortality exceeded 28 million ha [7]. In China, 14,100 hectares of Jilin province were affected by *I. typographus* in 2020 [8]. Similarly, *Ips grandicollis* (Eichhoff) caused millions of dollars of economic loss in 1994 in bushfire-damaged pine trees in Australia [9].

To cope with rapid bark beetle infestations and outbreaks, effective eco-friendly control methods such as semiochemicals are required [10]. Semiochemicals are the basis of communication in which chemical compounds released by an organism [11] modify the behaviour or physiology of another organism in a particular way [12], including to attract or repel the receiver. The semiochemical-based push–pull technique is a combination of an attractant and repellent [10], in which pests are repelled from the target site (push) while an attractant (pull) lures them away [13]. The pull component can be host volatiles—signalling food resources, habitat [14], breeding sites, or refugia [15]. For instance, *alpha* -pinene, found in pine trees and logs, is an attractive host cue for many bark beetles [16]. Alternatively, the pull component can be an aggregation pheromone imparting signals for feeding [12] and mate location [17]. Repellents are substances that induce an inhibitory response in insects. They can be plant-derived as host repellent or non-host volatiles (NHVs); e.g., limonene, linalool, benzaldehyde, and 4-allylanisole are known plant compounds used against bark beetles [18]. Insect-based repellents produced by insects in nature are used to avoid resource competition among broods and induce a repellent response in conspecific and heterospecific receiver species [19]. Verbenone is a well-known anti-aggregation pheromone for reducing *Dendroctonus* and *Ips* damage on host trees [20,21,22]. Moreover, some semiochemicals can be attractive at a low concentration and a repellent at a high concentration, or can induce synergistic effects in a mixture [10]. Trapping studies are of large potential interest as they are steps on the way towards finding the best semiochemical blends for tree protection. Previous studies using meta-analysis to quantify anti-attractant/repellent effects have examined only forest protection outcomes, such as on attack density or tree survival. The first such study [23] had a total of only 32 data points from papers published in 2000–2011 but still found an overall effect of Cohen’s d ca = –1 less tree attack, merged for two species, *D. ponderosae* and *I. typographus*. Raffa et al. [24] complemented this dataset with data from a multi-year study by Perkins, et al. [25]. They found 33 values were significantly below zero (39%), with 31 values having a value falling below zero (94%).

Here, we use meta-analysis [26] to aggregate and analyze data from 52 individual scientific studies to compare the overall effects of push–pull semiochemical treatments on reducing bark beetle (*Dendroctonus* and *Ips*) populations measured by a reduction in trap catches, tree mortality, and attacks on trees in different regions of the world. We also examine the effect of push treatment sources (conspecific, heterospecific, and NHVs) used alone or in combination. Specifically, we hypothesized that (1) the semiochemical push–pull treatment technique effectively reduces bark beetle populations and (2) the effectiveness of the semiochemical push–pull treatment varies between *Dendroctonus* and *Ips*.

## 2. Materials and Methods

### 2.1. Database Search and Selection Criteria

We used the Preferred Reporting Items for Systematic reviews and Meta-Analysis (PRISMA) protocol to collect relevant bibliography-sourced data for this study [27]. We selected two databases, Web of Science^®^ and SCOPUS^®^, from 1988 until 1 August 2022, using the following simple and standardized keywords arranged by Boolean Logic (“Semiochemical” OR “aggregation pheromone AND anti-aggregation pheromone” OR “attractant volatile AND repellent volatile” OR “attractant AND attractant disruptant” OR “attractant AND deterrent” OR “host volatile AND non-host volatile” OR “push pheromone AND pull pheromone”) AND (“*Ips*” OR “*Dendroctonus*”) [28].

### 2.2. Study Selection

Across both databases, 349 research articles were returned, of which 233 remained after the removal of duplicates, to which the following predefined eligibility criteria were applied:The study contained at least one species of *Dendroctonus* or *Ips* as the target.The study focused on the push–pull semiochemical technique; any study dealing with pull-alone, push-alone, insecticide use, and other control methods was excluded.The results were reported as means, variance (standard deviation or standard error), sample size, and other relevant statistical information to allow the calculation of the effect size.

Any study not meeting the above criteria was excluded from the analysis. Of the 233 research articles returned from the databases, 52 met these criteria (Figure 1). The selected research papers span 34 years, from 1988 to 2022 (Appendix A). We chose to also include trapping-only data points, as these provide a much larger sample and, thus, may show for many more species the potential for the best inhibitory blends to use in direct forest protection experiments.

### 2.3. Data Extraction

Metadata were extracted from the research papers, including sample size (number of replicates, n), means, and standard deviation (SD). If the data were presented as graphs, these were digitized using Web Plot Digitizer [29] and extracted as means and SD. When the standard error (SE) was reported in the studies, it was converted to standard deviation (SD) (SD = SE × √n).

This data extraction technique enhances the power of meta-analysis [30] and has been used in many previous meta-analyses [23]. Multiple studies from one research paper do not decrease independence or analytical power [31]; therefore, different semiochemical treatment–species variants from the same research article were considered as independent data. Parameters related to different species of the bark beetle genera *Dendroctonus* and *Ips*, treatment (compound and geographic regions), and response to treatment were collected from each study.

### 2.4. Meta-Analysis

To estimate the “push–pull” treatment effect of semiochemicals on reducing *Dendroctonus* and *Ips* population sizes compared to control, weighted mean of the log response ratio ($\bar{L*\;})$, also called ratio of means, was calculated [32]. We chose to use a newer effect size calculation better suited than the one used in the previous meta-analysis studies on tree experiment data [23,24] that describe effects in units of SD, Cohen’s d. Thus, data output is not the same, but significant data points are expected to be the same. The response ratio/ratio of mean (RoM) is largely unbiased, whereas the standardised mean difference, commonly known as Cohen’s d, may have non-negligible bias for small sample sizes [33] and some bias (about 5%) with sample sizes < 10. RoM also provides easier interpretation, which makes it a viable option for pooling data [34]. ($\bar{L*\;})$ was used to estimate effect size because it maintains symmetry for variables reported in different units through log transformation [35]. Moreover, calculating the percentage (%Δ) of effect size is simple from ($\bar{L*\;})$. Two components of variation in the sample log response ratios were calculated, sampling variation within Equation (1) and between Equation (2) experiments:(1)$$\upsilon =\frac{{\left(\mathrm{SDt}\right)}^{2}}{nt\bar{X}t^{2}}+\frac{{\left(\mathrm{SDc}\right)}^{2}}{nc\bar{X}c^{2}}\;$$

Equation (1): Standard deviation of treatment group (SD_t_), sample size/number of replicates in treatment group (nt), mean of treatment group ${\bar{X}}_{t}$, standard deviation of control group (SD_c_), sample size/number of replicates in control group (nc), and mean of control group ${\bar{X}}_{c}$.
(2)$${\hat{\sigma \;}}_{\lambda }^{2}=\frac{Q-\left(K-1\right)}{\;\sum_{i=1}^{k}W_{i}-\frac{\sum_{i=1}^{k}\;W_{i}^{2}{\;}_{\;}}{\sum_{i=1}^{k}W_{i}}}\;$$

In Equation (2), *K* represents the number of studies; *Q* represents the *Q* statistic (Equation (3)) used for testing the statistical significance of the between-experiment variance $\left({\hat{\sigma \;}}_{\lambda }^{2}\right)$ and is calculated by:(3)$$Q=\overset{k}{\underset{i=1}{\sum }}w_{i}{\left(L_{i}\right)}^{2}-\frac{{\left(\sum_{i=1}^{k}W_{i}\;L_{i}\;\right)}^{2}\;}{\sum_{i=1}^{k}w_{i}}\;$$

Here, in Equation (3), $w_{i}=1/\upsilon_{i}$ and ($L_{i}$) natural logarithm of the response ratio $L_{i}=ln\left({\bar{X}}_{t}\right)-ln\left({\bar{X}}_{c}\right)$

Effect size as weighted mean of the log response ratio ($\bar{L*\;})$ was calculated by Equation (4).
(4)$$\bar{L*\;}=\frac{\sum_{i=1}^{K}w_{i}^{*}L_{i}}{\sum_{i=1}^{K}w_{i}^{*}}\;$$

Here $w_{i}^{*}\;$is the reciprocal of the variance
wi*=1(υi+σ ^λ2)  

The effect size was expressed as a percentage (%Δ) and was calculated as:(5)$$\%\Delta =(e^{\bar{L*\;}}-1)\times 100$$
and standard error (*SE*) of this weighted mean ($\bar{L*\;})$ was calculated by Equation (6):(6)$$SE\left(\bar{L*\;}\right)=\sqrt{\frac{1}{\sum_{i=1}^{k}w_{i}^{*}}}$$

Lower (CI*_L_*) and upper (CI*_u_*) 95% confidence intervals for the mean log response ratio ($\bar{L*\;})$ were calculated using Equations (7) and (8), respectively:(7)$$95{\;\%\;\mathrm{CI}}_{L}=\;\bar{L*\;}-1.96\;SE\left(\bar{L*\;}\right)\;$$
(8)$$95{\;\%\;\mathrm{CI}}_{u}=\;\bar{L*\;}+1.96\;SE\left(\bar{L*\;}\right)\;$$

Significance level was computed using a two-tailed test. We calculated the pooled variances using the “escalc” function in the “metafor” package (version 2.4-0) in the R environment [36].

A heterogeneity test was performed before constructing the meta-analysis model to decide whether to use a fixed or random effect model. Cochran’s Q test of heterogeneity (Q) based on the full dataset (*k* = 863 observations) was highly significant (Cochran’s Q = 2.6 × 10^4^, df = 862, *p* < 0.0001) [36]. In the random effects meta-analysis, each study’s contribution to the results was weighted based on its contribution in the data synthesis. The inverse variance methods of “meta”[37] and “metafor” [36] packages in the R environment were used to assign the weights.

Forest plots were created with ggplot2 in the R environment to present estimated pooled effect size and their 95% confidence intervals (CI) produced by the meta-analysis [38]. If the 95% CI bar did not coincide with the zero line, or if the two-tailed test returned a *p*-value < 0.05, the effects of push–pull semiochemical treatment were considered significant [35]. Percentage of effect sizes is denoted by (±%), where a positive value shows an increase, and a negative value indicates a reduction in bark beetle population by the semiochemical deterrent treatment compared to the control.

The dataset was further categorized into parameters related to *Dendroctonus* and *Ips* species separately, source of test treatment (conspecific, heterospecific, NHVs, and host repellent either used alone or in combination), geographic region of an experiment (countries of an experiment where push–pull treatment was studied were coded/categorized as regions (Appendix A). To maintain heterogeneity in each observation, any variable reported in less than three studies (*k* < 3) was not included in the subgroup analysis.

## 3. Results

### 3.1. Metadata

Metadata were extracted from 52 research articles published in eight regions from 1988 to 2022. We obtained 863 observations (*k*) with push–pull semiochemical treatments using a uniform selection criterion. Most observations (72%) focused on reducing the population of nine *Dendroctonus* species, while the rest (28%) focused on 12 *Ips* species. Between-group heterogeneity existed in the sub-group analysis (Table 1).

### 3.2. The Overall Effect of Push–Pull Semiochemical Treatments on Dendroctonus and Ips

Our data synthesis showed a significant effect of push–pull semiochemical treatments on reducing bark beetle populations (*k* = 863). Significantly, semiochemical push–pull treatments reduced *Dendroctonus* and *Ips* population by 66% and 54%, respectively (Figure 2).

### 3.3. Effect of Push–Pull Semiochemical Treatment on Attack and Attraction of Dendroctonus and Ips Species

Of the total 624 studies identified on *Dendroctonus*, 506 aimed to reduce the attraction to lures in the presence of push semiochemicals, 152 focussed on the female and 153 on the male population, 103 studies examined reducing tree attacks, and 15 used the reductions in trees killed to assess efficacy (see Appendix A, Appendix A for details of response variables from the manuscripts).

On the other hand, of 239 total studies on *Ips,* 233 aimed to reduce *Ips* attraction to lures in the presence of push semiochemicals, 161 studies were with undefined sexes, 36 focused only on the female population, and 36 focused only on the male population. Six studies assessed push–pull semiochemical treatments to reduce tree colonization. This meta-analysis shows that push–pull semiochemical treatments significantly reduced attraction towards a lure in both the *Dendroctonus* and *Ips* population. Interestingly, in the presence of push semiochemicals, attraction to lures was reduced by 39% in *Dendroctonus* and 69% in *Ips*. In *Ips* species, the males, and in *Dendroctonus* species, the females serve as the pioneer attack sex by initially boring into the tree bark and attracting both sexes for a “mass attack”. A semiochemical deterrent treatment significantly reduced the frequency of the pioneer attack sex, in both *Dendroctonus* and *Ips*. The push-pull technique obtained a significant 81% reduction in trees killed by *Dendroctonus*, but non-significant reduction was obtained for *Ips* (Figure 3).

### 3.4. Effect of Push–Pull Semiochemical Treatments on Dendroctonus and Ips Species

The effect of push–pull semiochemical treatments varied significantly (*p* < 0.0001) for both the *Dendroctonus* and *Ips* species (Table 1). Notably, nine *Dendroctonus* species (*k* = 624) were examined in our meta-analysis; among them, eight species significantly reduced in population, while among 12 *Ips* species (*k* = 239), there was a significant population reduction in eight species by push–pull semiochemical treatments (Figure 4).

Two *Ips* species *I. perturbatus* and *I. latidens* show a 94% and 93% reduction in population by push–pull semiochemical treatments, respectively. Additionally, this semiochemical technique also showed a 86% reduction in the population of two *Ips* species, *I. avulsus* (Eichhoff) and *I. shangrila* (Cognato and Sun).

These results indicate an equivalent (69%) significant population reduction in *D. ponderosae* and *I. typographus*. In the studies considered where there was a reduction, the minimum was a 34% and 38% reduction for *D*. *rufipennis* (Kirby) and *Ips duplicatus* Sahlberg, respectively. These results demonstrate push–pull strategies’ potential effectiveness in reducing bark beetle populations, which cause significant damage to forests.

One *Dendroctonus* species, *D. terebrans* (Olivier), showed an increased population from push–pull semiochemical treatment. On the other hand, one *Dendroctonus* species (*D. mesoamericanus* Armendáriz-Toledano and Sullivan) and four *Ips* species (*Ips paraconfusus* (Lanier), *Ips subelongatus* (Motschulsky), *I. grandicollis,* and *I. mexicanus* (Hopkins)) did not show significant reductions with push–pull semiochemical treatments and highlight the need for further research (Figure 4).

### 3.5. Effect of Push–Pull Semiochemical Treatment Sources on Dendroctonus and Ips

We found seven treatments used alone or combined to protect softwood forests from *Dendroctonus* and *Ips* populations. Among them, six treatments were significantly effective for reducing both *Dendroctonus* and *Ips* populations (Figure 5).

Our results indicate that *Dendroctonus* and *Ips* have a 52% and 62% significant population reduction, respectively, when conspecific semiochemical treatment is used alone as a repellent.

The application of heterospecific semiochemicals is 69% effective for *Ips* and 20% for *Dendroctonus* when used alone as a repellent. Host repellent shows a 56% significant reduction in *Ips* but non-significant effects in *Dendroctonus*.

Interestingly, using push-pull semiochemical technique in combination provides an effective reduction in *Dendroctonus* and *Ips* population; for instance, conspecific semiochemicals combined with NHVs resulted in a 81% reduction in the *Dendroctonus* population and 77% in the *Ips* population; similarly, conspecific semiochemicals combined with heterospecific semiochemicals also result in a significant reduction in *Dendroctonus*, but population reduction was non-significant for *Ips*. These findings also indicate that, instead of using three semiochemicals, i.e., conspecific, heterospecific, and NHVs, their combination can provide an up to 92% reduction in the *Dendroctonus* population.

Heterospecific semiochemicals combined with host repellent were used only for *Ips* and provided a 59% reduction in this pest population.

### 3.6. Effect of Geographical Regions on Push–Pull Semiochemical Treatments in Dendroctonus and Ips

We examined data from five geographical regions for *Dendroctonus* and eight for *Ips* using a push–pull semiochemical treatment technique to reduce bark beetles’ population. Based on sub-group heterogeneity, the treatment efficacy varied significantly for *Dendroctonus* and *Ips* among geographical regions. In western Europe, Scandinavia, north-western North America, and eastern Europe, a 72–61% reduction in *Ips* population was obtained using push–pull semiochemical treatments. The implementation of push–pull semiochemical treatments in south-eastern North America resulted in a 75% reduction in *Ips* and 36% in the population of *Dendroctonus* (Figure 6). Similarly, in north-western North America, a 65% reduction in the *Ips* population and 48% in the *Dendroctonus* population is obtained using push–pull semiochemical treatments. There was a substantial decrease of 63% in the population of *Dendroctonus* in south-western North America; however, studies examining the push–pull effects in *Ips* in south-western North America and north-eastern North America have a small sample size and were not significant.

## 4. Discussion

The results of this meta-analysis study are consistent with the hypothesis that push–pull semiochemical methods can significantly reduce overall bark beetle (*Dendroctonus* and *Ips*) infestations. Although using the push–pull technique is challenging given the large size of forests, this technique has great potential to deter bark beetles. Indeed, soon after pheromone identification, scientists developed bark beetle control strategies using semiochemicals, particularly the push–pull techniques in experiments and field control [39].

Our findings align well with the more limited meta-analysis including only tree data for *D. ponderosae* and *I. typographus* [23] from 9 papers and 32 experiments from 2000 to 2011, where effects were moderate or strong for both species.

We also identified the different responses of the *Dendroctonus* and *Ips* species to push–pull semiochemical treatments. The same is true for the relationship between the treatment region and the bark beetle genus.

This meta-analysis found a significant reduction in the attraction of both male and female *Dendroctonus* and *Ips* beetles to the push-pull semiochemical treatments. Our results also significantly support the hypothesis of the success of push–pull semiochemical treatment in protecting trees from *Dendroctonus,* but a non-significant reduction in trees attacked by *Ips*, which could be due to the small number of *Ips* studies.

A significant reduction in the flight response of female *Dendroctonus* [40], tree attack rate [41], and tree mortality in *D. ponderosae* [42], and a reduced attraction to the aggregation pheromones in *D. pseudotsugae barragani* [43] and *D. valens* in the presence of push semiochemicals has been reported [44]. Likewise, an up to 60% reduction in the mortality of *Pinus albicaulis* by *Dendroctonus*, has been recorded in California using a push–pull technique [45]. These findings also indicate that semiochemical cues affect beetle orientation and flight capacity during flight. 

Push–pull semiochemical treatments significantly reduced the population of *Dendroctonus* species, with reductions ranging from 34 to 69%. This significant reduction in the population of the *Dendroctonus* species is aligns with the findings of Bedard, et al. [46]; Cook et al. [14]; Liu et al. [47]; Seybold et al. [41], Seybold and Fettig [48]; and Sullivan and Clarke [49] for *Dendroctonus brevicomis* (LeConte), *Dendroctonus pseudotsugae* (Hopkins), *D. valens*, *D. ponderosae*, *D. rufipennis*, and *D. frontalis,* respectively. Moreover, using *S*-(-)-verbenone in combination with 1-hexanol and (Z)-3-hexanol caused about a 60% reduction in *D. ponderosae* attacks on trees [45]. In summary, this ecofriendly technique is significantly effective at reducing the attraction and aggregation of *Dendroctonus* species toward lure/pull semiochemicals in the presence of repellent/push semiochemicals.

However, for *D*. *terebrans*, this meta-analysis shows significantly more attraction to pull semiochemicals even in the presence of push semiochemicals. Likewise, Sullivan et al. [50] found that *D. terebrans* attraction to lures frontalin, *exo*-brevicomin and an *alpha*-pinene was significantly increased by 4-allylanisole.

Our meta-analysis indicates up to a 94% population reduction for *Ips* species through push–pull semiochemical treatment. This meta-analysis found a 69% reduction in the population of *I. typographus*, a pest that killed many healthy *Picea* trees in Europe in 2007 [51]. A significant reduction in the attraction of this species to 2-methyl-3-buten-2-ol and (4S)-*cis*-verbenol was reported using *trans*-conophthorin and 1-hexanol [52], and by *trans*-verbenol in *I. shangrila* [53]. Similarly, a significant population reduction in *Ips pini* (Say) [54], *I. perturbatus* [55], *I. latidens* (LeConte) [56], and *I. avulsus* [57], and an up to 83% reduction in *Ips sexdentatus* (Boern.) [58] were reported using push–pull treatments.

Our analysis found a non-significant effect of push–pull treatments for *I. subelongatus,* and this result was similar to that reported by Yejing, et al. [59], who found no significant reduction in *I. subelongatus’* attraction to lure (S)-(−)-ipsenol and (S)-(+)-ipsdienol in the presence of NHVs such as myrtenol, (E)-3-hexen-1-ol, (E)-2-hexen-1-ol, and 1-hexanol, or by adding the anti-aggregation pheromone verbenone. Our findings for *I. paraconfusus* were inconsistent with Shea and Neustein [60], who reported that ipsdienol and verbenone successfully inhibited this bark beetle’s mass attack. The results of this metadata contradicted previous findings of Dickens, et al. [61] who found a significant reduction in *I. grandicollis’* attraction to lure ipsdienol, ipsenol, and *cis*-verbenol in the presence of the green leaf volatile hexanal. Similarly, Birgersson, et al. [62] found a significant reduction in trap catches of *I. grandicollis* in the presence of the heterospecific semiochemical lanierone. These results suggest that, although push–pull strategies might be useful for deterring populations of bark beetles like *I. grandicollis*, more research is required.

Except for the effect size of the host repellents, which is non-significant effect, our analysis supports our hypothesis that the populations of *Dendroctonus* spp. can be reduced by using various push–pull semiochemical treatment sources, either alone or in combination. Among the seven treatments used to reduce the *Ips* population, six significantly support our hypothesis, except for the non-significant effect of conspecific semiochemicals when combined with heterospecific ones. 

The findings of this metadata were aligned with the findings of previous research such as that of Sullivan [63], who reported a reduction in *Ips avulsus* catches toward ipsdienol and lanierone in the presence of *alpha*-pinene at 8 g/day. Wu, et al. [64] reported effective results in inducing repellence by NHVs from the bark and green leaves of angiosperms, and Cook et al. [14] in reducing aggregation in conifer-feeding beetles like *Dendroctonus* and *Ips* spp. on pine trees. Similarly, Zhang and Schlyter [18] reported an effective reduction in the aggregation of *Dendroctonus* spp. by using acetophenone, an NHV. Kandasamy, et al. [65] reported an effective reduction in the aggregation of *Dendroctonus* and *Ips* by benzyl alcohol, in combination with green leaf volatiles and other NHVs.

Jactel et al. [58] found up to an 83% reduction in attraction to aggregation pheromone by *trans*-conophthorin, whereas Dickens et al. [61] found the presence of 1-hexanol and hexanal effective for *Ips* deterrence and Fettig, et al. [66] reported 2-hexen-1-ol to significantly reduce the attraction of *Dendroctonus* spp. toward the lure in a trapping bioassay.

Conspecific semiochemicals can also act as anti-aggregants, e.g., Wu et al. [64] reported a repellent response in *D. frontalis*, *D. ponderosae*, *I. pini*, and *I. sexdentatus* to verbenone. Verbenone is a compound produced by oxidation of *alpha*-pinene by males of several *Dendroctonus* and *Ips* species as an anti-aggregation pheromone [67]. In field-trapping experiments, this semiochemical has been found to be effective in reducing the attraction of several bark beetles like D. *frontalis* toward lures [49] and *D. ponderosae* attraction to *trans*-verbenol, *cis*-verbenol, and *exo*-brevicomin [44].

*Trans*-verbenol, a conspecific anti-aggregation pheromone, has been found to significantly reduce the attraction of *Ips* and *Dendroctonus* toward aggregation pheromones [53]. Similarly, the presence of *trans*-verbenone and verbenone induces a push response in *Dendroctonus brevicomis* (LeConte) toward the pull semiochemicals (±) *exo*-brevicomin, (±) frontalin, and myrcene [46]. *Exo*-brevicomin is also a conspecific semiochemical that induced a repellent response in *D. valens* toward the aggregation pheromone [47]. Similarly, 3-methylcyclohex-2-en-1-one (MCH), a conspecific anti-aggregation pheromone, causes the reduction of attraction to 1-methyl-2-cyclohexen1-ol + frontalin + ethanol lures in *D. pseudotsugae barragani* [68].

Importantly, heterospecific semiochemicals effectively induce a deterrent response in bark beetles, as El-Ghany [15] reported that heterospecific semiochemicals serve as a chemical cue indicating that a competitor species already occupies a suitable host tree or stand. Similarly, Fettig et al. [55] reported that heterospecific semiochemicals, like *trans*-conophthorin, along with verbenone, is significantly effective in reducing the attraction to host trees.

Communication systems based on pheromones, like those found in bark beetle communities, play a vital role in insect interactions. The results of our meta-analysis show that the efficacy of NHVs to deter forest pests can be enhanced by incorporating the semiochemicals of a conspecific bark beetle; similarly, Seybold et al. [41] reported that using conspecific repellents with NHV reduced *Dendroctonus* and *Ips* attraction by signalling that early colonizers have previously attacked, and this is now an unsuitable host that should be avoided.

Crucially, it was observed that the combination of heterospecific semiochemicals with conspecific semiochemicals had a significantly stronger inhibitory effect in *Dendroctonus* than heterospecific semiochemicals alone, while the opposite was observed in *Ips*. Further confirmation came from Aukema and Raffa [69] and Seybold et al. [41], who found a greater pull response in *Ips* to frontalin, a pheromone of *D. frontalis*, but the exact reasons for the contradictory heterospecific effectiveness are unknown.

Our study demonstrates that semiochemical deterrent treatments are significantly effective for both *Dendroctonus* and *Ips* in various ecological conditions, and these results are aligned with those of earlier researchers. According to our findings the push-pull semiochemical technique was best in deterring *Dendroctonus* infestations in central-west North America by 70% and *Ips* infestations in south-eastern North America by 75%. In contrast, the non-significant effect of push–pull treatments from this meta-analysis for *Ips* in south-western North America (*p* > 0.05) and north-eastern North America (*p* > 0.05) found no support for these treatments as a deterrent, although, given the small sample size/studies, further research is needed. According to Fettig et al. [66], various studies conducted in north-western North America found acetophenone, (E)-2-hexen-1-ol + (Z)-2-hexen-1-ol, and verbenone provided significant protection to ponderosa pines from *D. brevicomis* attacks. Our results were aligned with the findings of Gaylord et al. [54] and Lindmark, et al. [70] for *I. pini* in south-western United States and *I. typographus* in Scandinavia, respectively.

For *Ips,* the population reduction in West Europe and East Europe lies between 72 and 61%, while, in East Asia, it is about 49%. It should be noted that NHV compounds naturally occurring in mixed habitats affect specialist herbivores, reducing their efficiency to locate their hosts. For effect size estimates of forest conditions on herbivore host location efficiency, see Jactel and Brockerhoff [71]. The semiochemical diversity hypothesis (SDH), suggested by Zhang and Schlyter [18], postulates a reduced searching efficiency of specialist herbivores in the face of NHVs, which may be one of the main factors for the reduction in herbivory [72] in mixed habitats.

This meta-analysis found that the effect of push-pull semiochemical treatments on *Dendroctonus* was highly significant and consistent with the findings of Sánchez-Martínez et al. [68] in north-western North America. It is worth noting that very limited datasets are available related to the effect of semiochemicals on *Ips*. Substantial geographical variation in pheromone response occurs between *Ips* populations [55]. Therefore, further research is required to explain the differences of the effects of semiochemicals on *Ips* reduction for different ecological regions.

This control technique has been used to reduce the devastation in forests caused by *I. paraconfusus* [14] and *D. brevicomis* [66], and to reduce the attraction of *Ips* toward lures [73], as well as protecting ponderosa pine, *Pinus ponderosa* var. *scopulorum* Engelm., trees from *Ips* attacks [54]. Like our findings, Lindmark et al. [70] also reported a reduction in bark beetle capture using an attractant–repellent approach and suggested using semiochemicals as an effective alternative to insecticides for reducing the bark beetle population.

## 5. Limitations and Suggestions

*Ips* spp. are recorded as serious pests in eight regions, while *Dendroctonus* is a pest in five regions of the world. Nine species of *Dendroctonus* and twelve *Ips* species pose a severe threat to the forest economy, but more studies focus on *Dendroctonus* than *Ips*. Therefore, more research is necessary to further comprehend the connection between the application of treatment sources and the reduction response in *Ips*.

The next wave of research on semiochemicals and bark beetles should take a systematic approach to other genera and examine the effects of the push–pull treatment and application mode (e.g., Specialized Pheromone & Lure Application Technology (SPLAT), aerial spray, and ground spread) on predators and parasitoids, which perform important roles in regulating bark beetle populations, at least in endemic ranges. Additionally, it is essential to quantify the impact of semiochemical push-alone and semiochemical pull-alone treatments on *Dendroctonus* and *Ips*.

## 6. Conclusions

The results of this meta-analysis indicate that the push–pull semiochemical technique can reduce *Dendroctonus* and *Ips* attraction to lures in forests. We conclude that (1)the use of the push–pull semiochemical technique is effective in reducing *Ips* (−66%) and *Dendroctonus* (−54%) compared to control; (2) among push–pull treatment sources, conspecific semiochemicals combined with heterospecific and NHVs provide the maximum reduction for *Dendroctonus*; (3) conspecific repellent semiochemicals can effectively lower the population of both genera by around half and, thus, can be used to protect the forest; (4) for *Ips,* heterospecific semiochemicals as the pull part of the technique is more effective when used alone rather than in combination; and (5) these deterrent techniques can diminish the pioneer attacking sex of both genera, even in the presence of an attractant bait, hence potentially halting future colonization and the attack density.

There is evidence that the push–pull semiochemical technique may be effective in preventing *Ips* attacks on softwood forests, but, to date, this research is quite limited; therefore, further research should evaluate the use of heterospecific semiochemicals with both NHVs and heterospecific semiochemicals in *Ips*. As part of ongoing *Ips* management plans, push–pull semiochemical treatments should be assessed, especially in native and invasive habitats.

## Figures and Tables

**Figure 1 insects-14-00812-f001:**
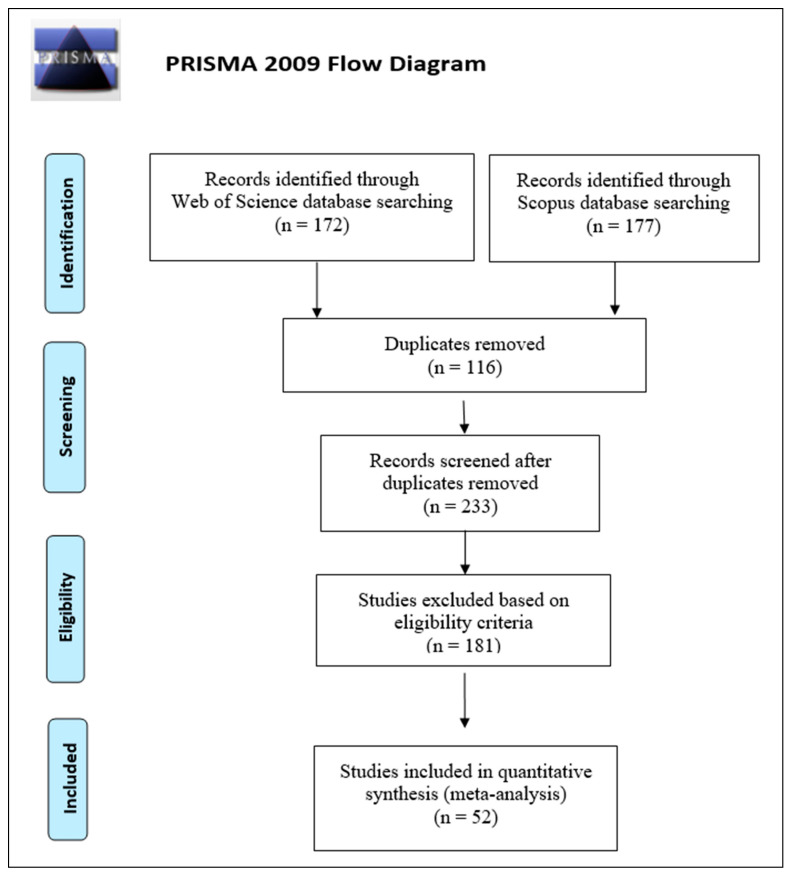
Preferred Reporting Items for Systematic reviews and Meta-Analysis flow diagram for meta-analysis. (After [27]).

**Figure 2 insects-14-00812-f002:**
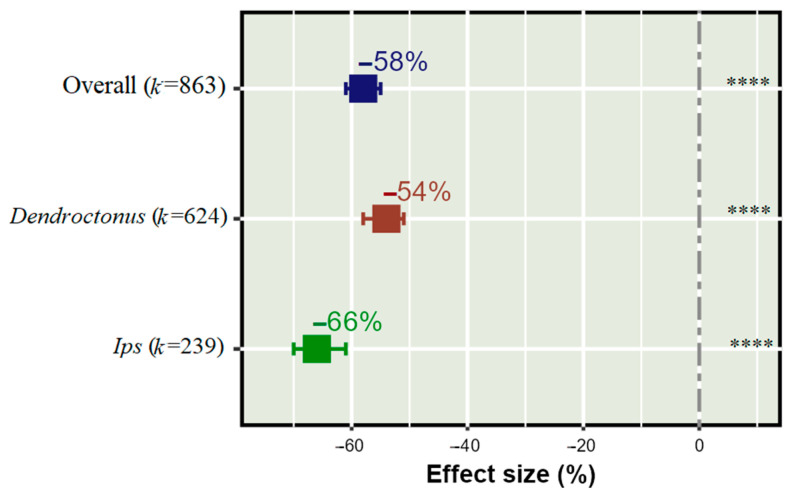
Overall effect of push–pull semiochemical treatments on reduction in bark beetle population. The error bars show 95% CI, and (*k*) indicates the number of studies for each variable. The significant level is denoted as **** for **** *p* < 0.0001 for the effect of treatment on bark beetle population.

**Figure 3 insects-14-00812-f003:**
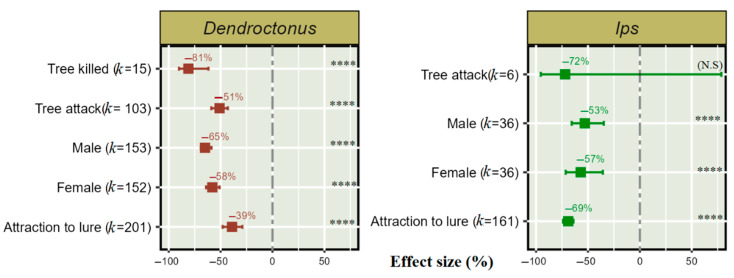
Effect of push–pull semiochemical treatment on the response of *Dendroctonus* and *Ips* populations. Error bar shows 95% CI, and (*k*) indicates the number of studies for each species. The significant level is denoted as **** for *p* < 0.0001 and “N. S” in parentheses indicates a non-significant effect of treatment on bark beetle population.

**Figure 4 insects-14-00812-f004:**
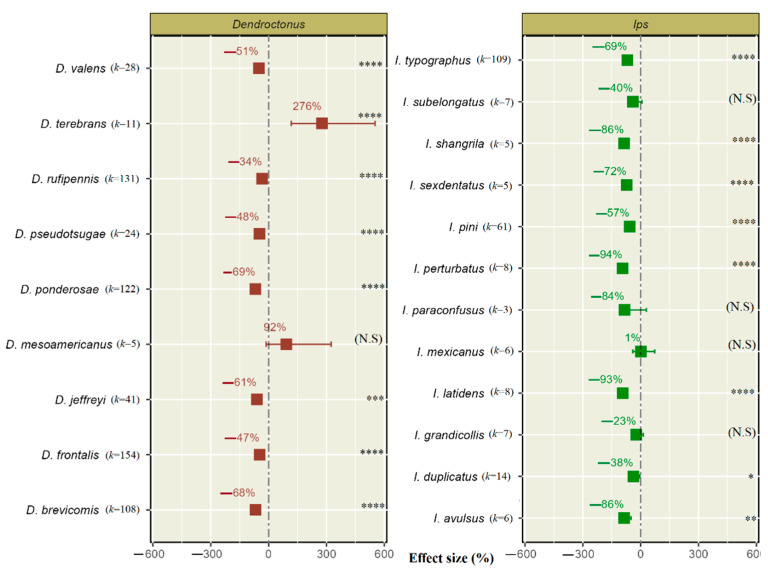
Effect of push–pull semiochemical treatments on reduction in *Dendroctonus* and *Ips* species population. The error bars show 95% CI, and (*k*) indicates the number of studies for each species. The significant levels are denoted as * for *p* < 0.05, ** for *p* < 0.01, *** for *p* < 0.001, and **** for *p* < 0.0001, and “N. S” in parentheses indicates a non-significant effect of treatment on bark beetle population.

**Figure 5 insects-14-00812-f005:**
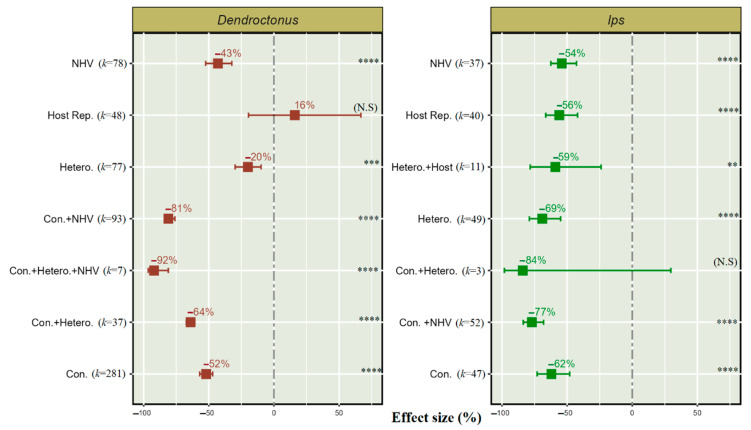
Effect of push–pull semiochemical treatment sources on the *Dendroctonus* and *Ips* population. Con. (conspecific), NHV (non-host volatiles), Hetero. (heterospecific), and Host Rep. (host repellent). Error bar shows 95% CI, and (*k*) indicates the number of studies for each treatment source. The significant levels are denoted as ** for *p* < 0.01, *** for *p* < 0.001, and **** for *p* < 0.0001, and “N. S” in parentheses indicates a non-significant effect of treatment on bark beetle population.

**Figure 6 insects-14-00812-f006:**
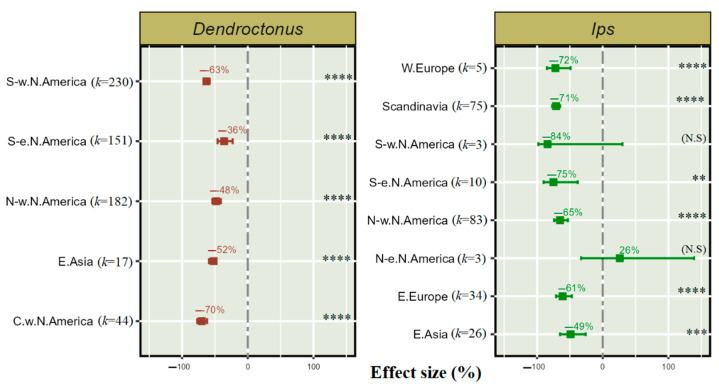
Effect of regions on push–pull semiochemical treatment for reducing *Dendroctonus* and *Ips* population among different geographical regions. The error bar shows 95% CI, and (*k*) indicates the number of studies for each region. The significant levels are denoted as ** for *p* < 0.01, *** for *p* < 0.001, and **** for *p* < 0.0001, and "N. S" in parentheses indicates a non-significant effect of treatment on bark beetle population.

**Table 1 insects-14-00812-t001:** Categorical variables and between-group heterogeneity (Q_b_) among observations (*k*) that affect *Dendroctonus* and *Ips* responses to push–pull semiochemical treatment.

Categorical Variable	*Dendroctonus*			*Ips*		
	*k*	Q_b_	*p*-Value	*k*	Q_b_	*p*-Value
Genus	624	-	-	239	-	-
Species	624	148.11	<0.0001	239	99.55	<0.0001
Treatment region	624	40.34	<0.0001	239	24.99	0.0008
Treatment Source	621	345.86	<0.0001	239	15.68	0.0282
Reduction (Repellence response)	624	32.41	<0.0001	239	6.89	0.075

## Data Availability

The data presented in this study are openly available online at Appendix A (selected publication references and metadata).

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
