# Peer review of "How Effective Are Push–Pull Semiochemicals as Deterrents for Bark Beetles? A Global Meta-Analysis of Thirty Years of Research"

_insects, 2023, doi:10.3390/insects14100812_

Round 1
Reviewer 1 Report
The TITLE & ABS are very good and exciting. Unfortunately, they are not well supported by content of paper. I was disappointed when coming to end of INTRO; end of 2nd to last para of INTRO –mid p2. See below!
In essence, this is a promising paper (or two papers: split in two parts; beetle behaviour vs beetle control (sensu stricto), below. However, much needs to be revised.
General data problem
Their big database presented have mostly data from trapping only with anti-attractants (field behaviour) but very little of control (recording lowered attacks or mortality from the studied insects). The data as presented do however not tell how much of data is simple trap tests (behaviour) and how much of the complex and expensive tree experiments (control s.s.). A glance at the huge table S1 suggest >>50% is just trapping?
Authors are correctly concerned that it is rarely possible to measure effects on population size [of a bark beetle] itself. To this end, they widen the scope of the paper, including various trapping tests, beyond correct understanding in the forest entomology.
For agriculture, field insects control direct ‘kill rates’ are possible, but due to the highly aggregated occurrence of bark beetles this is not reasonable –as noted. Most unfortunately, this this partially correct understanding leads to the inadequate inclusion as one of their effects measures the reduction of attraction of trap catches’. This a fatal flaw. Unfortunately, most of the material has seems to concern this type of effect (but I cannot find any exact number for this). Due to the large data set >800 rows of S1, but looking briefly at 1st and last page, there is >½ trapping data studies, at least if counting only row numbers.
The PRISMA 2009 flow diagram is a clear formal guide, but due to the misunderstanding of the biology of the system at hand by the authors, gives little help.
M&M section 2.2 studies included
On p 3, point 2 it is as well stated that studies including “push-alone” was excluded.
Again, this is not about control of a pest of field crops. Due to the specifics of both the semiochemical methods’ high costs for application as well as the value of the standing crop studies (mature trees), during high or medium (outbreaks) level populations provide the only ‘economic window’ for the testing of semiochemical type of insect control. In outbreak areas some presence of pheromone sources from attacked trees, monitoring or mass trapping will always exist.
I see a risk here authors may miss to include some of the most relevant studies?
Besides, production forest are not like a field crop with a 1 year crop rotation length. Tree rotation typically are of 50 -150 yeas in production forests. Thus, the standing crop is of immense value to protect for the forest management. In other words, the only relevant measure must be in (in ecological terms) the reduction of tree predation, quantified as tree attack (or density) or tree mortality.
GENERAL
1 Effects Measured: The inclusion of effect measured as reduction of trap catch.
Consider the following hypothetically example in a paper draft for a medical journal: “How effective are drugs of type X in preventing death by cardiac arrest? A meta-analysis”. However, the analysis turns out to have included as a majority studies that only measured HDL/LDL in blood (known to be related to cardiac disease), but missing to include any measures actual mortality in patients.
2 Missing Published Paper with Similar Method but citing Reference w/o Any Meta Data
Missing:
Schlyter, F. 2012: Semiochemical diversity in practice: anti-attractant semiochemicals reduce bark beetle attacks on standing trees—a first meta-analysis. Psyche: A Journal of Entomology 2012, Article ID 268621 | https://doi.org/10.1155/2012/268621.
(could be found in Google Scholar with search terms “bark beetle control meta”, 25 Citations).
Paradoxically,
Schlyter et al. 2015 is cited as a support for their meta-analysis procedure. This paper has no meta statistics whatsoever, only a comparison of pheromone components in Eurasian Ips spp in their Table 4. (Raffa et al. 2015 has some new meta data added to graph from S 2012.)
3 A New Effect Size Measure Less Pedagogically Introduced
The new measure used, could very well be a good one. Using the conventional “% reduction” could be good for the reader less used to effect size use. Still, new compared to Schlyter 2012.
A less math-oriented explanation why and how it differs in precision and accuracy to Cohen´s d used in the previous paper would be of interest. Even a few examples could be illustrative or to give compound measure of precision & accuracy or, in the main S1 table as a Cohen´s d column added. Alternatively, a plot of cases of % reduction compared Cohen´s d in studies here (k). Would it be possible to give simple formula for transferring the new measure from the old?
The regrssion of Cohen´s d on % Control is strong but not perfectly linear in S 2012 –a problem if field data variability of is there a mathematical reason?
PRESENTATION
Why are there only “tree plot” of only grouped cases not whole or part of material, that would for one thing show how many studies for each effect size measure or experiment type (trap /tree).
REFERENCES
Insects does not use DOI, why care must be taken to include vol, pp etc.
Surprisingly, many studies in mentioned in Tab S1 are not cited in REF of paper nor is some supplement form. I think this breaks basic meta-analysis rules?
Author Response
Thank you very much for taking the time to review this manuscript. Please find the detailed responses below and the corresponding revisions/corrections highlighted in the re-submitted files.
Comments and Suggestions for Authors
The TITLE & ABS are very good and exciting. Unfortunately, they are not well supported by content of paper. I was disappointed when coming to end of INTRO; end of 2nd to last para of INTRO –mid p2. See below!
In essence, this is a promising paper (or two papers: split in two parts; beetle behaviour vs beetle control (sensu stricto), below. However, much needs to be revised.
Revision made.
General data problem
Their big database presented have mostly data from trapping only with anti-attractants (field behaviour) but very little of control (recording lowered attacks or mortality from the studied insects). The data as presented do however not tell how much of data is simple trap tests (behaviour) and how much of the complex and expensive tree experiments (control s.s.). A glance at the huge table S1 suggest >>50% is just trapping?
Authors are correctly concerned that it is rarely possible to measure effects on population size [of a bark beetle] itself. To this end, they widen the scope of the paper, including various trapping tests, beyond correct understanding in the forest entomology.
For agriculture, field insects control direct ‘kill rates’ are possible, but due to the highly aggregated occurrence of bark beetles this is not reasonable –as noted. Most unfortunately, this this partially correct understanding leads to the inadequate inclusion as one of their effects measures the reduction of attraction of trap catches’. This a fatal flaw. Unfortunately, most of the material has seems to concern this type of effect (but I cannot find any exact number for this). Due to the large data set >800 rows of S1, but looking briefly at 1st and last page, there is >½ trapping data studies, at least if counting only row numbers.
The PRISMA 2009 flow diagram is a clear formal guide, but due to the misunderstanding of the biology of the system at hand by the authors, gives little help.
M&M section 2.2 studies included
On p 3, point 2 it is as well stated that studies including “push- alone” was excluded.
Again, this is not about control of a pest of field crops. Due to the specifics of both the semiochemical methods’ high costs for application as well as the value of the standing crop studies (mature trees), during high or medium (outbreaks) level populations provide the only ‘economic window’ for the testing of semiochemical type of insect control. In outbreak areas some presence of pheromone sources from attacked trees, monitoring or mass trapping will always exist. I see a risk here authors may miss to include some of the most relevant studies?
Besides, production forest are not like a field crop with a 1 year crop rotation length. Tree rotation typically are of 50 -150 yeas in production forests. Thus, the standing crop is of immense value to protect for the forest management. In other words, the only relevant measure must be in (in ecological terms) the reduction of tree predation, quantified as tree attack (or density) or tree mortality.
We have changed the title of the manuscript to more accurately reflect what we are testing. As detailed in the ms we have deliberately included “behaviour” (i.e. the deterrence of beetles from attractant traps as a proxy for a suitable/attractive host tree) rather than being restricted to the small number of studies using “control” with real trees. However, tree attack data are presented in Fig 3.
GENERAL
1 Effects Measured: The inclusion of effect measured as reduction of trap catch.
Consider the following hypothetically example in a paper draft for a medical journal: “How effective are drugs of type X in preventing death by cardiac arrest? A meta-analysis”.
However, the analysis turns out to have included as a majority studies that only measured HDL/LDL in blood (known to be related to cardiac disease), but missing to include any measures actual mortality in patients.
Thank you for pointing this, actual mortality/tree killed is presented in figure 3. We have changed the title of the paper from “control” sensu stricto (as measured by a reduction in tree mortality), to “How effective are push-pull semiochemicals as deterrents for bark beetles? A global meta-analysis of thirty years of research” to reflect the many studies we included in our analysis that used traps as an assessment method. We have mostly changed “control” to “deter” or “manage” throughout the ms to reflect this.
2 Missing Published Paper with Similar Method but citing Reference w/o Any Meta Data
Missing:
Schlyter, F. 2012: Semiochemical diversity in practice: anti- attractant semiochemicals reduce bark beetle attacks on standing trees—a first meta-analysis. Psyche: A Journal of Entomology 2012, Article ID 268621 | https://doi.org/10.1155/2012/268621.
(could be found in Google Scholar with search terms “bark beetle control meta”, 25 Citations).
Paradoxically,
Schlyter et al. 2015 is cited as a support for their meta- analysis procedure. This paper has no meta statistics whatsoever, only a comparison of pheromone components in Eurasian Ips spp in their Table 4. (Raffa et al. 2015 has some new meta data added to graph from S 2012.)
Thank you for pointing this – we made a typographical error and meant to cite the 2012 paper as in the references, we have corrected this.
3 A New Effect Size Measure Less Pedagogically Introduced
The new measure used, could very well be a good one. Using the conventional “% reduction” could be good for the reader
less used to effect size use. Still, new compared to Schlyter
2012.
A less math-oriented explanation why and how it differs in precision and accuracy to Cohen´s d used in the previous paper would be of interest. Even a few examples could be
illustrative or to give compound measure of precision & accuracy or, in the main S1 table as a Cohen´s d column added. Alternatively, a plot of cases of % reduction compared Cohen´s d in studies here (k). Would it be possible to give simple formula for transferring the new measure from the old?
The regrssion of Cohen´s d on % Control is strong but not perfectly linear in S 2012 –a problem if field data variability of is there a mathematical reason?
The response ratio/ratio of mean RoM is largely unbiased, whereas the standardised mean difference, commonly known as Cohen's d, revealed non-negligible bias for small sample sizes (Lin and Aloe, 2021) and some bias (about 5%) when there the sample size < 10. RoM also provides easier interpretation, which makes it a viable option for pooling data (Friedrich et al., 2008).
Lin L, Aloe AM. Evaluation of various estimators for standardized mean difference in meta-analysis. Stat Med. 2021 Jan 30;40(2):403-426. doi: 10.1002/sim.8781. Epub 2020 Nov 12. PMID: 33180373; PMCID: PMC7770064.
Friedrich, J.O., Adhikari, N.K. and Beyene, J., 2008. The ratio of means method as an alternative to mean differences for analyzing continuous outcome variables in meta-analysis: a simulation study. BMC Medical Research Methodology, 8(1), pp.1-15.
PRESENTATION
Why are there only “tree plot” of only grouped cases not whole or part of material, that would for one thing show how many studies for each effect size measure or experiment type (trap
/tree).
We have added a table to show the number of studies in each category.
REFERENCES
Insects does not use DOI, why care must be taken to include vol, pp etc.
Thank you for pointing this, we agree with your comment, and have corrected accordingly, the submission style to Insects is permitted to be in any style, and is corrected to the journal style on acceptance. A reference list of the selected 52 papers used in the meta-analysis has been added as a separate supplementary data file.
Reviewer 2 Report
I have the following specific comments below for authors' consideration:
1). Page 2, more details on "Insect-based repellents in nature" is it a insect itself produced or plant produced causing by insect's attack.
2). Page 8, authors mentioned about the softwood forests...How many softwoods and hardwoods among all analyzing samples? Are they significantly different? I did not see any discussion on hardwood pest management analysis. Is there any?
3). Page 9, five different geographical regions were further analyzed. In addition to the size, any other environmental factors were analyzed, or not, such as temperature or others?
Author Response
Thank you very much for taking the time to review this manuscript. Please find the detailed responses below and the corresponding revisions/corrections highlighted/in track changes in the re-submitted files
- Comment: Thank you for pointing this out. I agree with this comment. Therefore, I explained it in the text as the Insect-based repellents produced by insects in nature.
- Comment: The question about the softwood forests on Page 8, that how many softwoods and hardwoods were among all the analysing samples?
Response: Actually, all the data included in the paper pertains to softwood only.
- Comment: The five different geographical regions that were further analysed. In addition to the size, were any other environmental factors analysed, such as temperature or others on Page 9?
Response: No, other environmental factors were not analysed.
Reviewer 3 Report
This manuscript describes the use of meta-analysis on 30 years of data regarding the use of semiochemical push-pull methods to reduce and control the populations of bark beetles of the genus Dendroctonus and Ips. Among the conclusions were that “…the push-pull semiochemical technique can reduce Dendroctonus and Ips attraction to lures in forests.”
The statistical methods used to conduct this study seem sound, as do the conclusions drawn; overall the presentation is good, with several typos / editorial comments I noted below. Also, on my copy of the manuscript, there were no line numbers, also the authors’ names and affiliations weren’t listed.
-in the Introduction, it is mentioned that the Czech Republic experienced €260 billion in economic losses during the 2018-19 outbreak of Ips typographus (L.). This sounds like a very high amount; do the authors mean to say million, not billion?
-in the Introduction, italics are used frequently in different places, but may not be necessary.
-in the Introduction, “…of Jilin province, China,…” the word China doesn’t need to be repeated.
-after Equation 2, there should be a comma after “In equation 2”
-also, the second part of equation 4 is cut off / put down a line by the margin. This presentation error should be fixed.
-in the Results section and elsewhere, what is meant by “pioneer attack sex”? This should be explained.
-section 3.4, I would recommend rewording “…eight species significantly reduced in population.” To “…there was a significant population reduction in eight species.”
-also in this section, change “…and I. latidens shows…” to “…and I. latidens show…”
-Figure 4, in the caption, p<0.0001 should have 4 stars, not 3.
-in the Discussion, last sentence of the 4th paragraph, change “effect” to “affect”
-in the Discussion, 5th paragraph, is “Z3-hexenol” meant instead of “Z3-hexanol”?
-likewise, in the next paragraph of the discussion, exo-brevicomin is listed twice. I don’t think this is what the authors intended.
-in the Discussion, 10th paragraph, reword “…such as Sullivan (2023) reported…” to “…such as Sullivan (2023), who reported…”
-later in that same sentence, “…alpha-pinene at 8 g / day…” this sounds like a very high release rate; is this correct?
-in the next paragraph, I would recommend “…field trapping experiments” plural, not singular.
-in the paragraph after that, maybe capitalize “trans-Verbenol” as it is the first word of the sentence.
-same thing for “exo-Brevicomin” later in the paragraph.
-also in the same paragraph, state what “MCH” stands for; also in the same sentence, maybe change “…cause reduction to 1-methyl-2-cyclohexen-1-ol…” to “…causes reduction of attraction to 1-methyl-2-cyclohexen-1-ol…” I think this is what the authors intended to state.
-in the next paragraph, last sentence, rephrase to “…reducing attraction to host trees.” Instead of “…reduction attraction to host tree.”
-near the bottom of page 12, the second paragraph from the botton, “in” is repeated twice.
-in the Author Contributions section, clean up the punctuation.
There are no real issues with English language in this manuscript, except for typos and editorial comments I have included in my comments to authors.
Author Response
Dear Reviewer,
Thank you for dedicating time to review our manuscript. We appreciate your valuable feedback and have addressed your comments accordingly. Please see our responses below and find the corresponding revisions in the re-submitted files.
- In the Discussion, 5th paragraph, you mentioned "Z3-hexenol" instead of "Z3-hexanol."
Response: We appreciate your attention to detail, but the correct term in this context is "Z3-hexanol."
- The release rate of alpha-pinene being at 8 g/day, suggesting it may be a very high rate.
Response: We acknowledge your concern and would like to clarify that the release rate of alpha-pinene at 8 g/day is indeed accurate.
Once again, we thank you for your thoughtful review, and we hope that we make revission accordingly.
Best regards,
Round 2
Reviewer 1 Report
I selected the file name below, does not upload. I sent it to Ms Wu.
